# Isolation of *Candida* Species Is Associated with Comorbidities, Prolonged Mechanical Ventilation, and Treatment Outcomes in Surgical ICU Patients, a Cross-Sectional Study

**DOI:** 10.3390/jof10110743

**Published:** 2024-10-28

**Authors:** Josipa Glavaš Tahtler, Ana Cicvarić, Despoina Koulenti, Marios Karvouniaris, Maja Bogdan, Kristina Kralik, Irena Krajina Kmoniček, Marina Grbić Mlinarević, Slavica Kvolik

**Affiliations:** 1Department of Anesthesiology, Resuscitation and Intensive Care, Osijek University Hospital, 31000 Osijek, Croatia; ana.cicvaric@kbco.hr (A.C.); irena.krajina-kmonicek@kbco.hr (I.K.K.); marinagrbic147@gmail.com (M.G.M.); 2Medical Faculty, University of Osijek, 31000 Osijek, Croatia; maja.bogdan@kbco.hr (M.B.); kkralik@mefos.hr (K.K.); 32nd Critical Care Department, Attikon University Hospital, 15772 Athens, Greece; despoina.koulenti@nhs.net; 4UQ Centre for Clinical Research, Faculty of Medicine, The University of Queensland, Brisbane, QLD 4029, Australia; 5Intensive Care Unit, AHEPA University Hospital, 54636 Thessaloniki, Greece; karvmarevg@hotmail.com; 6Department of Clinical Microbiology and Hospital Infections, Osijek University Hospital, 31000 Osijek, Croatia

**Keywords:** *Candida albicans*, non-albicans candida, invasive candidiasis, mechanical ventilation, intensive care unit, *Candida* species, surgical patients, outcomes

## Abstract

The isolation of *Candida* may be related to comorbidity, prolonged mechanical ventilation, and survival during intensive care unit (ICU) stay, especially with non-albicans *Candida* (NAC). To examine the frequency of *Candida* isolation, associated comorbidities and outcomes in the surgical ICU in Osijek University Hospital, Croatia, the data from the electronic database from May 2016 to 30 June 2023 were analyzed. In a cross-sectional study examining 15,790 microbiological samples, different strains of *Candida* were observed in 581 samples from 236 patients. The control group (N = 261) was 130 consecutive patients from March to May 2019 and 131 in the same months in 2020 (pre- and post-COVID-19). Comorbidities, duration of mechanical ventilation, and survival were compared. Patients with isolated *Candida* were more often non-elective and had significantly more heart, kidney, and liver diseases and sepsis than the control group (*p* < 0.001). The duration of mechanical ventilation was 9.2 [2.2–9.24], 96 [24–146], 160 [19.5–343], and 224 [73.5–510] hours in the controls, in patients with *Candida albicans*, in patients with NAC, and in patients with ≥2 *Candida* species isolated, respectively. The mortality was significantly higher (42%) in patients with isolated *Candida* than in the control group (19%, *p* < 0.001). In a multivariate analysis adjusted for patients’ age, the Simplified Acute Physiology Score II, days of ICU, and type of admission, only sepsis on admission was an independent predictor of mortality (odds ratio = 2.27).

## 1. Introduction

Fungal infections are common in critically ill patients. Candidiasis is a fungal infection caused by the *Candida* species. Candidiasis can present with different clinical manifestations, including bloodstream infections (candidemia), mucosal infections (e.g., thrush and vaginal yeast infections), and invasive candidiasis affecting internal organs. Factors contributing to invasive fungal infections are a damaged skin barrier or mucous membrane, and the incidence is higher in immunosuppressed and neutropenic patients [1]. Other factors favor the development of invasive candidiasis, the most important of which is the long-term use of broad-spectrum antibiotics, older age, a high SOFA score, diabetes mellitus, severe hepatic failure, and septic shock [2]. According to a retrospective cohort study by Gouel-Cheron and coworkers, long-term mechanical ventilation, the use of multi-lumen catheters, extracorporeal membrane oxygenation, renal replacement therapy, long-term operations, the use of immunosuppressive drugs, and antineoplastic agents were also associated with invasive candidiasis [1,3].

Infections with candidas are responsible for increasing morbidity and mortality, leading to prolonged treatment in intensive care units (ICUs) and extended hospital stays [4,5,6]. *Candida* species cause the most fungal infections in intensive care units, and they account for 70–90% of invasive fungal infections, with a total mortality of ~40–60% [4,5,6]. Candidiasis is the seventh most frequent cause of nosocomial infections, with an incidence of 7.07 per 1000 ICU admissions [2,3].

Most cases of invasive candidiasis occur from the 5th to the 12th day of treatment in intensive care units [7]. Invasive infections are more frequent when candidas are isolated from stool and urine samples and in multifocal colonization [5]. Blood culture is the gold standard for diagnosing invasive candidiasis, but the sensitivity varies between 21% and 71% due to inadequate sampling or deep-seated candidiasis [8].

Candidas can cause infections of numerous organ systems [9]. The clinical presentation is nonspecific. Candidemia can cause sepsis and septic shock, and the dissemination of candidas can also occur and cause infections of the lungs, eyes, liver, spleen, central nervous system, heart, and heart valves, as well as other organs and tissues, causing abscesses [5,6]. Intra-abdominal candidiasis most often occurs due to perforations, anastomotic leak, relaparotomy, necrotizing pancreatitis, and transplantation of abdominal organs, and has a higher incidence in surgical ICUs [1].

Although they are closely related, it is very important to distinguish between colonization and *Candida* infection [5]. *Candida* colonization refers to the presence of *Candida* species on mucosal surfaces or in body fluids without infection. With ICU patients, we often deal with possible *Candida* carriers, and some species are part of the physiological microbiota of the upper respiratory, gastrointestinal, or urogenital system. *Candida* colonization is present in 5–15% of patients admitted to the ICU, although some authors have reported that 70% of patients were already colonized on admission to the ICU [3,10]. Patients are asymptomatic but may be potential reservoirs for infection transmission and candidiasis development. During extended stays in the ICU, the number of patients colonized with *Candida* increases to 50–80%, of which 5–30% develop invasive candidiasis [5].

The most common cause of fungal colonization and infection is *C. albicans* strains. In recent years, there has been an increase in *non-albicans Candida* (NAC), causing approximately 40 to 50% of infections [9,11]. The NAC species include *C. glabrata*, *C. krusei*, *C. tropicalis*, and *C. parapsilosis* [6,7]. A weaker response to azole therapy is associated with NAC [6,7,9].

Patients who are admitted to surgical ICUs, in addition to those with surgical diseases, often have numerous other comorbid conditions that can favor *Candida* colonization and the development of candidiasis. Such diseases certainly include hypertension, diabetes, and malignant diseases. In surgical patients, comorbidity can also affect treatment outcomes and lead to prolonged ventilation and higher mortality.

This study aimed to examine the frequency of *Candida* spp. isolation in surgical ICU patients and its association with comorbidities, emergency versus elective admission to the ICU, and inflammatory biomarkers on ICU admission. Furthermore, we wanted to analyze whether *Candida* isolation is associated with the duration of mechanical ventilation, days of ICU stay, and mortality.

## 2. Materials and Methods

We conducted a cross-sectional study and analyzed data from patients admitted to the intensive care unit, Osijek University Hospital, Croatia, from 2016 to 2023. The study group included adult patients admitted to the ICU in whom *Candida* was isolated, regardless of the type of sample, the number of yeasts, or whether it was infection or colonization (Figure 1). This is because samples for microbiological analysis are only routinely taken from patients who have had clinical, laboratory, or radiologically confirmed signs of infection. The control group was patients without positive *Candida* findings in microbiological isolates. The control group was formed by subtracting from the electronic database all consecutive patients admitted to the surgical intensive care unit during the three-month period in the year before the start of the COVID-19 pandemic from March to the end of May 2019 (N = 160), and the same number of patients after the start of the COVID-19 pandemic, from March to the end of June 2020 (N = 161). A total of 557 patients were analyzed, of which 236 (47.5%) had isolated *Candida* and 231 (52.5%) were in the control group (Figure 1).

Demographic and clinical data were collected from an electronic database and medical charts. The following data were analyzed: age, sex, type of admission (emergency, elective), the surgical department from which the patient was admitted to the ICU, days of ICU stay, type of surgical procedures (craniotomy, laparotomy, thoracotomy), reoperations, initial laboratory findings taken on admission to the ICU, SAPS II score within 24 h of ICU admission, the duration of mechanical ventilation, SAPS II score 24 h before discharge or death, and period of hospitalization (before and during the COVID-19 pandemic). The treatment outcomes analyzed were duration of mechanical ventilation (hours), ICU stay (days), and ICU survival.

### 2.1. Comorbidities

Comorbidities on admission were analyzed separately. Heart diseases included atrial fibrillation, previous myocardial infarction, unstable angina, cardiomyopathy, and valvular diseases. Arterial hypertension, atherosclerosis, and peripheral vascular diseases were classified as vascular diseases. Respiratory diseases included asthma, chronic obstructive pulmonary disease, pulmonary fibrosis, and previous tuberculosis. Any previously recognized disease, such as Alzheimer’s disease, Parkinson’s disease, personal history of a stroke, and conditions with a new disorder of the consciousness such as brain trauma, were considered neurological diseases. The patient, who had a brain tumor with impaired consciousness, was classified as having a neoplasm and a neurological disease. Coagulopathy was considered any condition in which the patient had a coagulation disorder requiring therapeutic intervention upon admission, i.e., thrombophilia, previous therapeutic or preventative use of anticoagulants, or history of deep venous thrombosis or pulmonary embolism. Trauma and polytrauma were evaluated separately. Renal diseases were considered when acute and chronic kidney failure was confirmed according to KDIGO criteria and kidney tumors. Bladder diseases such as infections, stones, or tumors were classified as urological diseases. Gastrointestinal diseases known before admission, such as tumors, Crohn’s disease, ulcers of the stomach, and duodenum were registered, whereas hepatobiliary diseases were a separate category. Due to the high mortality rate, sepsis on admission with any confirmed type of microorganisms in the blood cultures was considered as an entity, regardless of the source of infection [5]. Diabetes and hyperlipidemia were classified as metabolic diseases, while hypo/hyperthyroidism, parathyroid gland diseases, and pituitary gland diseases were considered endocrine diseases. A psychiatric diagnosis was recorded when the patient or their guardian declared that they had the specified diagnosis and/or was taking psychoactive drugs. Soft tissue infections, i.e., neck abscesses, mediastinitis, Fournier gangrene, etc., were considered as specific groups. Neoplasms were considered both in patients whose cancer surgery was the main reason for surgery and admission to the ICU and in those who had trauma and other tumors that were previously or currently under therapy, such as prostate cancer.

### 2.2. Microbiological Analyses

Microbiological samples were taken when infection was suspected based on clinical or laboratory signs. Depending on the possible source of infection urine, blood, vascular catheter tips, drains, and wound swabs were collected according to standard guidelines. If the patient had multiple intravascular accesses, samples for blood cultures were taken from all of them. Respiratory specimens were tracheal aspirates and bronchoalveolar lavage (BAL) which were taken during bronchoscopy. All urine culture samples were taken via a urinary catheter. Analyses of the samples received were performed by routine microbiology procedures that account for inoculation onto selective and nonselective agar plates, identification of morphologically different colonies with germ tube test, API Candida© (BioMerieux, Marcy-l’Etoile, France), and MALDI TOF ultrafleXtreme Biotyper, Bruker, Ettlingen, Germany. Film Array BCID2 Panel was performed if blastoconidia were observed from positive blood culture samples inoculated in BACTEC BD© bottles (Becton Dickenson, Wokingham Berkshire, UK). Susceptibility of identified isolates was performed using ATB Fungus 3© (BioMerieux, Marcy-l’Étoile, France) and interpreted according to valid EUCAST standards.

### 2.3. Statistical Analysis

Categorical data are represented by absolute and relative frequencies. Numerical data are described by the median and the limits of the interquartile range. Differences in categorical variables were tested with the χ^2^ test. Differences in numerical variables between two independent groups were tested with the Mann–Whitney *U* test, whereas differences between three or more groups were tested using one-way ANOVA. Differences in variables between the two measurements were tested with the Wilcoxon test. The correlation between variables was examined with Pearson’s correlation for parametric variables (*r*) and Spearman’s correlation (ρ) for non-parametric variables. Factors that were significant in bivariate analysis were tested in multivariate analysis. A *p* < 0.05 was considered statistically significant.

## 3. Results

The database for the ICU Department at Osijek University Hospital, Croatia, includes the results of microbiological samples taken between 22 May 2016 and 30 June 2023. During that time, a total of 5118 patients were admitted to the ICU, and 15,790 samples were taken for microbiological analysis. Pathogenic microorganisms were confirmed in 4971 samples, and the presence of *Candidas* was confirmed in 581 isolates (11.7% of all positive samples) taken from 236 patients. *C. albicans* was confirmed in 426 (73%) isolates, followed by *C. glabrata* (53, 9.1%), *C. parapsilosis* (45, 7.7%), *C. tropicalis* (13, 2.2%), and *C. kefyr* (7, 1.2%). *C. rugosa*, *C. famata*, *C. krusei*, and *C. lusitaniae* were registered in four patients (0.7%); *C. guillermondi* in two (0.3); and *C. metapsilosis* and *C. orthopsilosis* in one patient. Their azole resistance is shown in Appendix A.

*Candida* isolation was most often confirmed in urine cultures, followed by tracheal aspirates (27 and 18/1000 admissions). The overall incidence of *Candida* isolation was 46 cases per 1000 admissions. Invasive candidiasis was registered in 54 patients (1.06% of all ICU admissions). The characteristics of these isolates are shown in Table 1.

As shown in Table 1, the largest number of *Candida* species was isolated from urine cultures (58.5%), followed by tracheal aspirates (39.8%) and blood cultures (13.1%). *C. albicans* was isolated in most patients (68.2%), while NAC was isolated in 75 (31.8%) of all patients. A significant increase in the frequency of NAC was observed within a period after the COVID-19 pandemic (chi-square test *p* = 0.009). In 13.6% of the patients, two or more candidas were isolated, and in 30.5% of the patients, in addition to candidas, various pathogenic bacteria were also isolated (Table 1). The isolation of candidas from ≥2 samples was associated with the presence of pathogenic bacteria in microbiological samples (Spearman ρ = 0.177, *p* =0.007). Candidemia was observed in 31 patients, with NAC found in 17 (55%) positive blood cultures.

The majority of patients in *Candida* group and the control ICU population were men (Table 2).

Although non-elective patients predominated among all ICU patients, the proportion of non-elective patients was significantly higher in the population with *Candida* isolation (*p* < 0.001). In patients with *Candida* isolation, mortality was significantly higher at 42%, while in the control group, it was 19% (*p* < 0.001). There were no differences in ICU mortality between patients colonized with *C. albicans* and those colonized with NAC (*p* = 0.908).

The most common comorbidities were vascular diseases, and arterial hypertension was the most common diagnosis. Among the patients in the *Candida* group, there were significantly more heart diseases, respiratory diseases, renal diseases, coagulopathies, hepatobiliary, sepsis, soft tissue infections, and metabolic diseases, whereas neoplasms were less frequent than in the control population of surgical ICU patients (Appendix A).

The frequency of candidemia was higher in women and was associated only with a higher SAPS II score at admission within that group (Spearman’s correlation ρ = 0.13, *p* = 0.046).

The presence of *Candida* isolation between 22 May 2016 and 30 June 2023 was registered more often after abdominal surgical procedures, with the highest frequency occurring after reoperation, as shown in Appendix A. Also, *Candida* isolation occurred more often in hospitalized patients with nosocomial bacterial infections. In patients with nosocomial infections, the frequency of *Candida* isolation from multiple samples was also higher (Spearman’s rank correlation, ρ = 0.177, *p* = 0.007), as was the frequency of *Candida* isolation from the abdominal cavity (ρ = 172, *p* = 0.008). NAC significantly correlated with diabetes (ρ = 0.192, *p* = 0.003) and abdominal procedures (ρ = 0.2, *p* = 0.002).

The patients with *Candida* isolation had a significantly prolonged ICU stay, higher median values of CRP and PCT at admission to the ICU, and a higher SAPS II score within 24 h of ICU admission and before discharge than control patients (Table 3).

The duration of mechanical ventilation in patients with multiple samples that were positive for *Candida* was longer compared to that of patients in whom *Candida* was isolated from one sample only, with a median of 71 (22–214) hours vs. 244 (76–246) hours (one-way ANOVA, *p* < 0.001) (Appendix A). The duration of mechanical ventilation depends on the type of isolated *Candida.* The average duration of mechanical ventilation was 103 h in patients with isolated *C. albicans*, 160 h in patients with NAC species, and 224 h for patients with two or more isolated species of *Candidas* (one-way ANOVA, all *p* ≤ 0.05 vs. control) (Figure 2).

In a bivariate analysis, the association of individual risk factors with mortality was examined for the whole population. The highest OR (odds ratio) for mortality was confirmed for non-elective admission, isolation of *Candida* vs. the control group, and heart and renal diseases (Table 4).

Risk factors that were significantly associated with mortality in bivariate analysis were tested in multivariate analysis and adjusted for age, SAPS II score, days of ICU, and type of admission (classified as elective or non-elective). The model was statistically significant as a whole, (χ^2^ = 389.5, *p* < 0.001), and it explains between 55% (according to Cox and Snell) and 78% (according to Nagelkerke) of the variance of the negative outcome (death) and exactly classifies 92% of cases. Only one predictor in the multivariate analysis was significant, and that was the presence of sepsis on admission (OR = 2.27) (Appendix A). The same results were obtained in the logistic regression performed within the group with *Candida* isolation, where sepsis was also the most important predictor of mortality (Appendix A).

## 4. Discussion

This study confirmed that *Candida* species isolation is related to the length of mechanical ventilation. The isolation of *Candida* species is also associated with heart, renal and metabolic comorbidities, with more non-elective admissions and increased mortality in the group of surgical patients admitted to the ICU.

**The duration of mechanical ventilation** is significantly higher in our patients with *Candida* isolation than in the control population. Similar results were confirmed by Huang et al. in a meta-analysis that included 1661 mechanically ventilated adult patients in the ICU. They compared all those who were positive with *Candida* with those who were negative and excluded immunocompromised patients [12]. In their study, the *Candida*-colonized population had a higher 28-day mortality (RR, 1.64; 95% CI, 1.27–2.12), although probably due to the higher mortality, the duration of treatment in the ICU was not different between groups [12]. According to the results of published studies, NAC is often isolated in patients who had previous antifungal therapy, diabetes on SGLT-2 therapy, and those who were immunocompromised [13,14,15]. In our study, prolonged mechanical ventilation was associated with the presence of any *Candida*, and especially with NAC. Considering that NAC is the most frequent cause of sepsis in our patients, their impaired condition caused by sepsis may result in prolonged mechanical ventilation. Other associated comorbidities such as diabetes, which was also associated with NAC in our study, may contribute to prolonged mechanical ventilation.

**High mortality** in patients with candidiasis has been observed in other studies and meta-analyses. In a large meta-analysis including 10,692 patients with candidemia, Zhang et al. recorded a mortality of 49.3% (95% CI 45.0% to 53.5%) and significantly prolonged ICU stay [6]. High mortality in patients with candidiasis was also found in other studies. According to the results of the EUCANDICU project, mortality in invasive candidiasis was 42% [2]. Aguilar et al. confirmed a mortality of 33% in 1149 surgical patients with invasive candidiasis, while the mortality in the total population of surgical patients in the ICU was 13%. All cases of invasive candidiasis were confirmed in patients with peritonitis and severe sepsis or septic shock [11]. The total frequency of invasive candidiasis was 19.1 per 1000 stays, similar to our study where we recorded 18.3 per 1000 ICU admissions, respectively. The authors confirmed the presence of central venous catheters, urinary catheters, antibiotic therapy, mechanical ventilation, prolonged stay in the ICU and total parenteral nutrition as risk factors [6,11,16]. Apart from total parenteral nutrition, these risk factors are present in most patients in the ICU.

Most of these studies did not analyze the patients’ previous comorbidities but the SOFA score as an indicator of organ dysfunction at the time of admission to the ICU. In our study, we analyzed the diagnoses of patients upon arrival in the ICU. This study, which covers a longer period and a larger number of patients, confirmed that both *Candida* isolation and mortality are associated with patient comorbidities. The most significant indicators of higher frequency of candidiasis are heart, metabolic and respiratory diseases, and coagulopathy.

**Non-elective patients** represent about 90% of our patients with *Candida* isolation, and they are mainly admitted after emergency abdominal surgeries, commonly in septic shock. *Candida* in the abdominal cavity is considered a serious infection that requires immediate treatment [2,17]. Bassetti et al. in their research proved that the most common form of invasive candidiasis are candidemia and intra-abdominal candidiasis, with a frequency of about 5% at admission to the intensive care unit, while the 30-day mortality was 42% [2]. Risk factors for increased mortality include extremes of age, severe liver failure, septic shock, and increased SOFA score. We did not observe significant differences in age between the groups. This is because we included only the adult population. Furthermore, elderly patients, who are often very frail, are usually treated with more conservative methods, and only a few of them were in our surgical population. In our patients with *Candida* isolation, there were 90% non-elective, i.e., emergency patients. The majority were admitted to ICU with peritonitis, their SAPS II score was significantly higher, as well as the frequency of heart and respiratory diseases, coagulopathy, and sepsis. Sepsis was associated with soft tissue infections such as Fournier’s gangrene, abdominal and pelvic abscesses. In the multivariate analysis, sepsis was confirmed as the most important factor that contributed to mortality in our entire population. Candidemia and septic shock with increased mortality were associated with NAC in our patients, which was also reported in other studies [18]. Routsi et al. observed the incidence of candidemia was 10.2% during COVID-19, which is significantly higher compared to the two pre-pandemic periods when the incidence of candidemia was 3.2% and 4.2%, respectively [19]. Comparable to the results of other authors, an increase in fluconazole resistant strains was also observed in our patients in the post-COVID-19 era [19,20].

**Heart diseases**, among which atrial fibrillation is dominant, are significantly more common in our population with *Candida* isolation than in the control group of all surgical ICU patients. Previous atrial fibrillation is rarely cited as a risk factor for candidiasis in surgical patients. Other factors that can contribute to more frequent candidiasis in heart patients are their significant hemodynamic instability, the need for an invasive approach, catheterization of central veins and arteries. Candidemia is a frequent cause of infective endocarditis and new-onset atrial fibrillation [21]. A series of studies confirmed the association of *Candida* with the presence of foreign bodies which are commonly used in patients with atrial fibrillation, such as central venous catheters, intraarterial catheters, urinary catheters, endotracheal tubes, or parenteral nutrition [11,13]. Therefore, one of the possible ways to reduce candidiasis could be the application of the ERAS protocol in emergency patients, who comprise most of our population. These measures are faster removal of the nasogastric tube and drains, reduction of catheterization time, earlier mobilization, and earlier initiation of patient feeding [22].

**In diabetic patients**, who have a higher risk of being admitted to the ICU from the emergency room, the application of the ERAS protocol could be particularly significant [23]. In our population with candidiasis, there are twice as many of them as in the control group. New therapies such as sodium-glucose co-transporter 2 inhibitors (SGLT2), which inhibit renal glucose reabsorption and cause glucosuria, favor candida colonization and candiduria [14,24]. Due to the increased frequency of heart disease and diabetes in emergency patients, early recognition of this risk group, application of the ERAS protocol, differentiation of colonization from infection and early initiation of therapy could reduce poor outcomes of these patients [23,25].

**Respiratory diseases** were significantly associated with *Candida* isolation in our study. This is consistent with the observation of Krause et al. that *Candida* was never isolated from the lungs of healthy patients [26]. They compared isolates obtained from patients undergoing elective plastic surgery after intubation, who did not receive antibiotic therapy, with a group that was mechanically ventilated without antibiotic therapy and with a group that was not mechanically ventilated but received antibiotics for other infections. They confirmed that the presence of *Candida* is frequent from respiratory specimens in ICU patients who were mechanically ventilated, or who received antibiotics, and that it reflects lower respiratory tract dysbiosis [26,27]. The diagnosis of *Candida* pneumonia is demanding because the clinical presentation and radiological findings are nonspecific, and the *Candida* isolate should be interpreted with caution and individually for each patient [27,28].

Some authors believe that *Candida* is a marker of a greater burden of illness, and that it is not clear whether it should be actively treated or is only an indicator of immunosuppression in patients [20,29,30,31] Although the guidelines of professional societies do not recommend routine therapy in patients with colonization of the lower respiratory tract, most of them recommend that every clinical case with *Candida* isolation should be considered in the context of the clinical presentation [31,32]. These newer recommendations are based on the results of clinical observations and preclinical studies [33,34]. In a group of 200 ICU patients in whom *Candida* was confirmed in lower respiratory tract samples, Lanigan et al. observed a mortality rate of 80% [34]. The authors confirmed that antifungal treatment was associated with better outcomes, and that mortality was higher in patients who did not receive antifungal therapy [34].

**Candiduria** in asymptomatic patients without risk factors may be treated by changing the urinary catheter, while in high-risk patients, fluconazole is necessary to prevent invasive candidiasis [33,34,35]. According to Dias, in critically ill ICU predisposed inpatients with numerous comorbidities, any candiduria, asymptomatic or symptomatic, may be the only clue of a severe infection [35]. Catheterization of the urinary bladder is the most common risk factor for candidiasis in patients in ICUs, followed by use of antibiotics, prolonged hospital stay, extreme age, diabetes mellitus, female gender, immunosuppressive therapy [36,37,38]. We expected that urine cultures would be positive more often in diabetics than in other patients with *Candida*, which was not confirmed (Spearman’s correlation coefficient ρ = 0.002, *p* = 0.972). Marchena-Gomez et al. confirmed that candidiasis is not only related to mechanically ventilated patients in the ICU but is common in general surgery wards. The frequency of *Candida* is like that in our study, but in their patients, *Candida* was not the most common in urine cultures (18.6%) but is highest in surgical wounds in which the frequency of isolation was 25% [39]. Anemia was an independent factor of mortality in their study [39].

In our population of surgical ICU patients with *Candida* isolation, there is a significant predominance of *C. albicans*, while NAC is observed in 32% of patients. Other studies have confirmed similar ratios, but a slightly higher frequency of NAC. In the study by Dimopoulos et al. 35.7% of NAC was found, as well as 41% in that of the authors Aguilar and coworkers [11,13]. According to the results of recent studies, the frequency of fluconazole-resistant NAC strains has increased since the COVID-19 pandemic, as observed in our study. This increase in resistant strains is probably related to the longer use of antibiotics and the more invasive diagnostic and therapeutic procedures. Therefore, the importance of reducing antibiotic use, preventing fungal colonization, early diagnosis of infections, and the need to use scoring systems is emphasized. These measures can facilitate the differentiation of patients at low and high risk of infection and should be combined with diagnostic tests to select patients for no treatment or those who are candidates for preventive treatment with confirmed fungal isolation [13,40].

**The limitation** of our study is that, except for invasive candidiasis, we did not associate the isolated *Candida* species with the signs of clinical infection. It is, therefore, likely that, as described in other studies, some of the isolates represented colonization that may be associated with other bacterial or viral infection [41]. Samples for microbiological analysis were taken only from patients with laboratory and/or clinical signs of infection. Therefore, we believe that the isolation of *Candida* species is a consequence of the patient’s general impaired condition and previous comorbidities, as these were found to be significantly different between patients with candida and the control group in this study. However, only sepsis on admission was found to be an independent predictor of mortality.

Another limitation of this study is its retrospective character. Since the aim of the study was to compare any *Candida* isolation with the duration of mechanical ventilation and comorbidities, we did not analyze the antibiotics and other medications the patients received before taking the samples. Since most of the patients were admitted to the ICU with complicated infections, respiratory failure, or septic shock, a relatively high frequency of administration of these drugs was expected. Earlier studies confirmed an increased rate of *Candida* infection in patients who received antibiotics for a long time and the association of *Candida* isolation with corticosteroid therapy [6,11], so we believe that this was also the case in our patients. The association of all these factors could be better examined in a future prospective study.

## 5. Conclusions

In conclusion, this study confirmed the association of any *Candida* isolation in surgical ICU patients with unfavorable outcomes. These adverse outcomes are prolonged ICU stay, prolonged mechanical ventilation, which is dependent on the type of *Candida* isolated, and increased mortality. Considering the high frequency of non-elective patients with multiple comorbid diseases in populations with candidas, special attention should be paid to septic patients, the possibility of reducing invasive procedures, earlier mobilization, earlier diagnosis, and determining which patients will benefit from empirical or targeted antifungal therapy.

## Figures and Tables

**Figure 1 jof-10-00743-f001:**
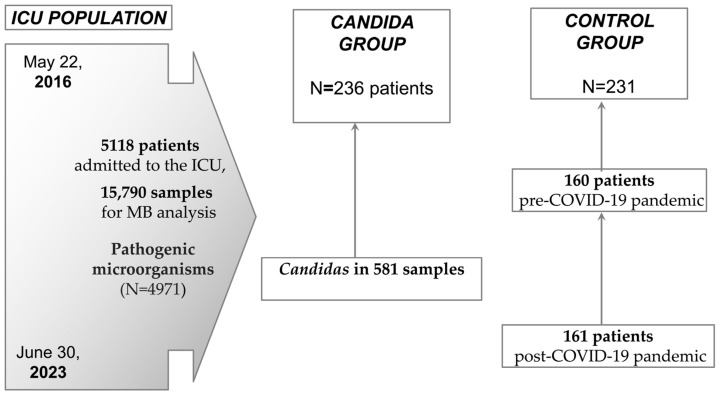
Patient flow in the period between 22 May 2016 and 30 June 2023. ICU—intensive care unit, MB—microbiology.

**Figure 2 jof-10-00743-f002:**
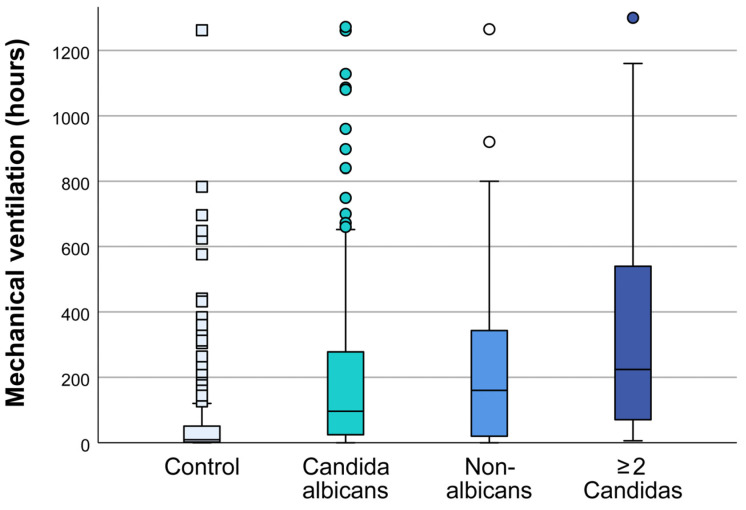
Mechanical ventilation in the surgical intensive care unit (ICU) patients according to the species of *Candida* isolated. Boxplots present medians and interquartile ranges of mechanical ventilation in a control group of ICU patients and in patients with *Candida* isolation during their ICU stay. Dots indicate outliers.

**Table 1 jof-10-00743-t001:** Characteristics of samples in 236 patients with confirmed *Candida* isolation during the ICU treatment in the period from May 2016 to June 2023.

All Patients with *Candida*		N = 236 (%)
Type of positive isolate	Urine cultures	138 (58.5)
	Tracheal aspirates	94 (39.8)
	Blood cultures	31 (13.1)
	Swabs from thoracic drains, abdominal cavity, cerebrospinal fluid	29 (12.3)
	Wound cultures	18 (7.6)
	Surveillance swabs	1 (0.4)
Total number of positive samples	1 sample	137 (58.3)
	≥2 samples	98 (41.7)
*Candida* species (all patients)	*C. albicans*	161 (68.2)
	*Non-albicans*	43 (18.2)
	≥2 *Candidas*	32 (13.6)
Pathogenic bacteria associated with *Candida* at the time of sampling		72 (30.5)
Pre-COVID-19 vs. post-COVID-19 * patients		154:82
	*Non-albicans*	40 (25.9):35 (42.7)

Number (%) of samples with *Candida* isolation. Some patients had ≥2 positive samples. * The period up to 29 February 2020 was considered as pre-COVID-19 and the period after 1 March as the post-COVID-19 period.

**Table 2 jof-10-00743-t002:** A comparison of gender, type of admission, and treatment outcomes in patients with isolated *Candida* and control ICU patients.

		*Candida* (N = 236)	Control (N = 261)	Total	*p*
**Age (years)**		66 (57–75)	65 (58–73)		0.46 ***
**Gender**	Male	132 (56)	169 (65)	301 (61)	0.05 ^†^
	Female	104 (44)	92 (35)	196 (39)	
**Admission to the ICU**	Elective	23 (10)	116 (44)	139 (28)	**<0.001** ^†^
	Non-elective	213 (90)	145 (56)	358 (72)	
**Outcome**	Death	99 (42)	49 (19)	148 (30)	**<0.001** ^†^
	Discharged	134 (58)	212 (81)	346 (70)	

* Mann–Whitney *U* test, medians, and interquartile ranges are shown. ^†^ For each categorical variable, a number (%) of patients is shown, and the χ^2^ test was calculated to confirm differences between patients with isolated *Candida* and the control group of ICU patients. Statistically significant differences are bolded.

**Table 3 jof-10-00743-t003:** Laboratory findings and SAPS II score in surgical patients with *Candida* isolation during the ICU treatment and in the control group of subsequent surgical ICU patients between 22 May 2016 and 30 June 2023.

	*Candida* (N = 236)	Control (N = 261)	Difference	95% CI of Difference	*p **
ICU days	9 (3–19)	3 (2–6)	−5	−6 do −3	**<0.001**
SAPS II on admission	56 (43–68)	45 (35–58)	−10	−13 do −6	**<0.001**
SAPS II at discharge	49 (30–79)	33(26–45)	−12	−16 do −7	**<0.001**
WBCs on admission (×10^9^/L)	14.3 (9.9–20.8)	13.4 (9.9–17.2)	−0.9	−2.1 do 0.3	0.14
CRP on admission [mg/L]	162 (74–245)	73 (20–168)	−65	−90 do −43	**<0.001**
PCT on admission [mg/L]	2.9 (1.3–17.8)	1.3 (0.5–7.6)	−0.9	−2.1 do −0.1	0.03

* Mann–Whitney *U* test, medians, and interquartile ranges are shown. ICU—intensive care unit, SAPS—Simplified Acute Physiology Score. WBCs—white blood cells, CRP—*c*-reactive protein, PCT—procalcitonin. Statistically significant differences are bolded.

**Table 4 jof-10-00743-t004:** Bivariate regression analyzing the relationship of individual risk factors with mortality in the whole population of patients with *Candida* isolation and in the control group of surgical patients.

Risk Factors	ß	OR	95% CI	*p*
Elective or non-elective admission	1.52	4.55	2.59–7.99	**<0.001**
Age	0.018	1.02	1.004–1.03	**0.01**
SAPS II on admission	0.043	1.04	1.03–1.06	**<0.001**
SAPS II at discharge	0.124	1.13	1.11–1.16	**<0.001**
Isolation of *Candida* vs. control	1.16	3.20	2.13–4.97	**<0.001**
Reoperation	0.631	1.88	1.26–2.81	**0.002**
Mechanical ventilation (hours)	0.001	1.001	1.001–1.002	**<0.001**
Days of ICU	0.017	1.02	1.004–1.03	**0.01**
Comorbidities				
Heart diseases	0.961	2.60	1.765–3.86	**<0.001**
Vascular diseases	0.196	1.22	0.81–1.82	0.34
Respiratory diseases	0.481	1.62	1.09–2.38	**0.02**
Coagulopathy	0.673	1.96	1.11–3.47	**0.02**
Poly/trauma	0.002	1.002	0.58–1.74	0.99
Neurological diseases	0.165	1.18	0.80–1.75	0.41
Renal diseases	0.771	2.16	1.42–3.29	**<0.001**
Gastrointestinal diseases	0.214	1.24	0.84–1.83	0.28
Hepatobiliary diseases	0.653	1.92	1.20–3.08	**0.007**
Sepsis	1.54	4.67	3.06–7.14	**<0.001**
Neoplasms	−0.95	0.39	0.25–0.59	**<0.001**
Metabolic diseases	0.510	1.67	1.09–2.53	**0.02**
Endocrine diseases	0.143	1.15	0.75–1.78	0.52
Psychiatric diseases	−0.077	0.93	0.56–1.52	0.76

SAPS—Simplified Acute Physiology Score. Statistically significant differences are bolded.

## Data Availability

The raw data supporting the conclusions of this article will be made available by the authors on request.

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
