# Peer review of "Isolation of Candida Species Is Associated with Comorbidities, Prolonged Mechanical Ventilation, and Treatment Outcomes in Surgical ICU Patients, a Cross-Sectional Study"

_jof, 2024, doi:10.3390/jof10110743_

Round 1
Reviewer 1 Report
In general terms work have inherent bias since is considered the presence of Candida spp., in urine, traqueal aspirate and swabs, these samples are not recommended as samples to associate Candida spp with the infections, besides is not a microorganism followed in thes kind of samples.
Thus, the bias associated are several.
There is not relation between a proper design of control and cases (more controls than cases).
The use of proper smples that really represent an association between microorganism and disease (no swabs)
Differences between colonizaton and real role as pathogen.
Author Response
|
Response to Reviewer 1 Comments
|
|||
|
1. Summary |
|
|
|
|
Thank you for dedicating time to review this manuscript. Below, please find the detailed responses and the associated revisions or corrections, which are highlighted or tracked in the changes of the resubmitted files.
|
|||
|
2. Questions for General Evaluation Does the title describe the article's topic with sufficient precision?
Does the introduction provide a comprehensive yet concise overview about the state of knowledge in the area of research?
|
Reviewer’s Evaluation
No. With the title is and induction to have a look about the role of Candida as agent assotiated with pneumonia
Yes. |
Response and Revisions Thank you for your comment. We appreciate your opinion that the title does not fully reflect the paper's main point. Although the title clearly highlights key aspects of the study, ensuring that potential readers can immediately recognize the article's relevance to the field, we have recognized that the Candida species, the type of admission, and comorbidities may also be emphasized. However, the most important finding was that the type of Candida was associated with the duration of mechanical ventilation. We have changed our title accordingly. If you feel that other findings should be prioritized, we are open to additional suggestions that could improve the title. |
|
|
Is the research design appropriate and are the methods adequately described? |
No. There is a lack of rigour in methods, this is a typical control and cases and controls, in this design controls most be doubled in comparison with cases, in this cases are almos equal Then. the use of swabs as sample is not relevant since it is not able to descrimine between colonizant and as a possible pathogen. Comparison between previous swabs (as Candida score, Ostrozky score or another one) was not determined. Then a huge of bias are related.
|
Thank you for pointing this out. We understand that the number of patients could have been higher and that the ratio of controls to cases was not ideal, but the number of control patients refers to the average population of all patients in the intensive care unit, and even if there were twice the number of control patients, the result would be the same. |
|
|
|
|
|
|
|
Are the results presented clearly and in sufficient detail, are the conclusions supported by the results and are they put into context within the existing literature?
Are all of the cited references relevant to the research?
Does this article provide a relevant contribution to the scientific discussion of this topic?
English language and style |
No. As samples, and relationship between cases an controls was not controlled rigourosly, association can not be made. The use of association and regresion as statistic were not realized.
Yes.
Yes.
Extensive editing of English language required. |
We appreciate your comment. While we acknowledge that the controls-to-cases ratio may be different, it should be noted that the number of control patients represents the average population of all patients in the intensive care unit. According to all measured outcomes—mechanical ventilation, ICU days, and mortality—the outcomes of patients with isolated candida are at least twice as bad as those of the control population. Increasing the control group would not change this result. Our biostatistician did both correlation, bivariate, and multivariate regression analyses of data.
Thank you for the comment about the need for extensive editing of the English language. We have thoroughly revised the manuscript to improve clarity, grammar, and general readability. All corrections are shown in the track changes.
|
|
|
3. Point-by-point response to Comments and Suggestions for Authors |
|||
|
Major comments: In general terms work have inherent bias since is considered the presence of Candida spp., in urine, traqueal aspirate and swabs, these samples are not recommended as samples to associate Candida spp with the infections, besides is not a microorganism followed in thes kind of samples. Thus, the bias associated are several.
Response 1: Dear Reviewer 1. Thank you for pointing this out. We agree with this comment. Therefore, we have discussed the limitations of using urine samples and tracheal aspirates to link Candida spp. to infection in the manuscript. Only one surveillance swab was analyzed. This was a surveillance swab of the nasopharynx in a patient with a skull base fracture and cerebrospinal fluid leak. I must reiterate that the aim of this work was not to examine the frequency and type of microorganisms that caused the infections. We presented these infectious agents in another paper (https://pubmed.ncbi.nlm.nih.gov/39203352/). Our and other studies have confirmed that Candida isolation is observed in sicker patients and is associated with multiple comorbidities and emergency procedures. In this work, we wanted to see if candida isolation is related to prolonged mechanical ventilation or some specific diseases. We thought that diabetes could be the most significant here. In the correlation analysis, it was shown that the isolation of NAC is associated with diabetes, which we have shown in the results. A comparison of the control group and the group with Candida isolation was made using both a chi-square test and a correlation analysis. Candida was associated with heart, respiratory, hepatobiliary, renal, and metabolic diseases, coagulopathies, sepsis, and soft tissue infections. The results were confirmed both by chi-square and correlation analysis tests. Candida was also associated with nosocomial infections.
|
|||
|
|
|||
|
Detail comments: There is not relation between a proper design of control and cases (more controls than cases). The use of proper smples that really represent an association between microorganism and disease (no swabs) Differences between colonizaton and real role as pathogen. |
|||
|
Response 2: As we stated in the previous answer, the aim of the work was not to connect the isolation of candida with the infection. We have already noted that we believe that the isolation of candida is a consequence of the patient's general impaired condition and previous comorbidities. Therefore, we analyzed patients' comorbidities and confirmed significant differences between patients with candida and control groups. However, only sepsis on admission was found to be an independent predictor of mortality. Thank you again for your suggestions. |
|||
Reviewer 2 Report
General Impression
The authors present a well-written manuscript about a study on the association of Candida colonization/infection with outcomes in surgical patients in a Croatian Intensive Care Unit. The authors open with a comprehensive and accurate review of the current literature on candidiasis in the hospital setting. The study is a cross-sectional retrospective review of patient charts, and the statistical analysis of patient data is appropriate and well-described. Data are presented in compelling and easy to understand format, and the inclusion of supplementary data is helpful to researchers engaged in similar projects. The breakdown of factors for morbidity and mortality by multivariate regression analysis is useful for detailing and quantifying the effects of Candida on health outcomes. Sample size is sufficient and the statistical treatment of data by non-parametric methods is appropriate. The authors find a correlation between Candida isolation and length of ICU stay, duration of mechanical ventilation and mortality. This interpretation is supported by the data and should be of interest to the readers of the Journal of Fungi. The limitations of the study – lack of data about antibiotic and corticosteroid treatment before admission – are inevitable and are discussed in the proper context.
Detailed and specific comments
The manuscript has been carefully written and contains no logical flaws. Even after repeated reading, I have no reasonable suggestions for improvement. My apologies for not being able to contribute constructive criticism.
Author Response
Dear Reviewer 2
Thank you for your comments. We did a considerable amount of work and are happy you like it.
Reviewer 3 Report
Dear editor and authors!
The article „The impact of Candida isolation on the duration of mechanical ventilation and outcomes in surgical ICU patients, a cross-sectional study” concerns the analysis of data regarding de facto 236 patients in the field of Candida. Data analysis was conducted since 2016. The article is valuable in terms of collecting data that may remain unnoticed in everyday practice.
I think the article is of interest to International Journal of Fungi readers.
The article is interesting, but requires major changes in the layout of the texts and the clarity of the presented analyses, so I recommend a minor revision.
Abstract:
18: I would only recommend mentioning the number of samples shown for the Candida genus and mentioning the number of patients. (581 samples from 236 patients).
1.Introduction:
-the introduction requires correction
· It should be mentioned that we often deal with possible carriers, some species are part of the physiological flora.
· What factors predispose to the development of Candidiasis ? (move to the beginning from 45 and supplement with other relevant factors)
· In my opinion, the analysis of the patient age factor in the above context is missing. Because it does not give the full view of the problem.
· What is the reason for low candidaemia detection?
· Paragraph 63-69 requires moving up.
· 72: “Candida colonization is present in 5-15%” - is definitely higher up 50-86%
In eg. https://doi.org/10.3390%2Fijms12107038
2. Materials and Methods
· A graphical diagram of data classification would be very useful for the reader
· The description of this part is huge and unclear. It needs to be simplified
· Microbiological analysis should be detailed in a paragraph and heading.
Results:
TIPS:
· Where possible, I would supplement the presented data with Candida species. (Fig 1)
· Captions, require improvement of italics.
Discussion:
· It is extensive due to the analysis of many factors, but I have no objections.
Strong points of the work:
Extensive data analysis
Statistical analysis
Weak points of the work:
Introduction requires editing
Data methodology that requires simplification or simple presentation to the reader
In my opinion, the article is suitable for publication, but requires corrections.
Regards,
R
Author Response
|
Response to Reviewer 3 Comments
|
|
||
|
1. Summary |
|
|
|
|
Thank you very much for taking the time to review this manuscript. Please find the detailed responses below and the corresponding revisions/corrections highlighted/in track changes in the re-submitted files
|
|
||
|
2. Questions for General Evaluation |
Reviewer’s Evaluation |
Response and Revisions |
|
|
Does the introduction provide sufficient background and include all relevant references? |
- the introduction requires correction • It should be mentioned that we often deal with possible carriers, some species are part of the physiological flora. • What factors predispose to the development of Candidiasis ? (move to the beginning from 45 and supplement with other relevant factors) •In my opinion, the analysis of the patient age factor in the above context is missing. Because it does not give the full view of the problem. • What is the reason for low candidaemia detection? • Paragraph 63-69 requires moving up. • 72: “Candida colonization is present in 5-15%” - is definitely higher up 50-86% In eg. https://doi.org/10.3390%2Fijms12107038 |
Candida colonization is discussed in ln. 66-68. We have changed this as you suggested: Factors contributing to invasive fungal infections are a damaged skin barrier or mucous membrane, and the incidence is higher in immunosuppressed and neutropenic patients (1). Other factors favor the development of invasive candidiasis, the most important of which is the long-term use of broad-spectrum antibiotics, older age, and high SOFA score, diabetes mellitus, severe hepatic failure, and septic shock Thank you for your observation regarding age. We did not discuss age extremes, both neonates and the very elderly, which are at special risk for candidiasis since they may be specific risk categories. Their characteristics – bacterial infections, the use of antibiotics and immunosuppression are mentioned as risk factors. We have mentioned reasons for low candidaemia detection – deap-seated candidiasis and inadequate sampling. We did the changes as you suggested and added reference https://doi.org/10.3390%2Fijms12107038 |
|
|
Are all the cited references relevant to the research? |
Yes/Can be improved/Must be improved/Not applicable |
|
|
|
Is the research design appropriate? |
A graphical diagram of data classification would be very useful for the reader • The description of this part is huge and unclear. It needs to be simplified • Microbiological analysis should be detailed in a paragraph and heading. |
As you suggested we have done a graphical diagram showing in a simplified manner how the groups were made.
We created new subheadings and separated the microbiological analysis into a separate paragraph. We have added new details related to microbiological analysis. |
|
|
Are the methods adequately described? |
Yes/Can be improved/Must be improved/Not applicable |
|
|
|
Are the results clearly presented? |
Yes/Can be improved/Must be improved/Not applicable |
|
|
|
Are the conclusions supported by the results? |
Yes/Can be improved/Must be improved/Not applicable |
|
|
|
3. Point-by-point response to Comments and Suggestions for Authors |
|
||
|
Comments 1: I would only recommend mentioning the number of samples shown for the Candida genus and mentioning the number of patients. (581 samples from 236 patients).]
|
|
||
|
Response 1: Thank you for pointing this out. We put the number of samples and patients with confirmed Candida isolation into the Abstract.
|
|
||
|
Comments 2: 1.Introduction: -the introduction requires correction should be mentioned that we often deal with possible carriers, some species are part of the physiological flora.
· What factors predispose to the development of Candidiasis ? (move to the beginning from 45 and supplement with other relevant factors)
· In my opinion, the analysis of the patient age factor in the above context is missing. Because it does not give the full view of the problem.
· What is the reason for low candidaemia detection?
· Paragraph 63-69 requires moving up.
· 72: “Candida colonization is present in 5-15%” - is definitely higher up 50-86% In eg. https://doi.org/10.3390%2Fijms12107038 |
|
||
|
Response 2: Agree. We have, accordingly, revised Introduction section to emphasize this point. We added a sentence about age extremes, that were not included in our study. Neonatal patients were not included in our study, and we had no very old patients, who usually do not undergo extensive surgical procedures. A mean age of our population is 66 (57 – 75) in Candida, and 65(58 – 73) in the Control group. We have changed all the points that you have suggested, and added thew following sentences: With ICU patients, we often deal with possible Candida carriers, and some species are part of the physiological flora of the upper respiratory, gastrointestinal, or urogenital system. Candida colonization is present in 5-15% of patients admitted to the ICU, although some authors reported that 70% of patients were already colonized on admission into the ICU. We have added a reference you suggested. We have moved ln. 63-69 up. We have explained reasons for candidaemia detection - due to inadequate sampling or deep-seated candidiasis and added a new reference here. We have discussed age of our population, whisc was almost the same in both groups in discussion: In our patients, we did not observe significant differences in age between the groups. This is because we included only the adult population. Furthermore, elderly patients, who are often very frail, are usually treated with more conservative methods, and there were few of them in our surgical population. 2. Materials and Methods · A graphical diagram of data classification would be very useful for the reader. Thank you, we did it as you suggested. · The description of this part is huge and unclear. It needs to be simplified. Thank you, by adding a flow chart it can be easily read. · Microbiological analysis should be detailed in a paragraph and heading. Thank you, for this suggestion. We have added new subheadings and several details on microbiological samples were taken. Microbiological samples were taken when infection was suspected based on clinical or laboratory signs. … If the patient had multiple intravascular accesses, samples for blood cultures were taken from all of them. Respiratory specimens were tracheal aspirates and bronchoalveolar lavage (BAL) which were taken during bronchoscopy. All urine culture samples were taken via a urinary catheter. Results: TIPS: · Where possible, I would supplement the presented data with Candida species. (Fig 1). Thank you for this observation. We have listed all Candida species in ln. 247-250, and in Table S1. Since there are only a few of some species, for further analyses, we have grouped them into Candida albicans and non-albicans groups, as suggested by our microbiologist, prof. Maja Bogdan. Captions, require improvement of italics. Thank you, we re-checked italics as you suggested in whole manuscript. Weak points of the work: Introduction requires editing – Thank you, as you suggested we have changed an Introduction section, added two new references, and additional changes that are singned in blue in the revised version od the manuscript. Data methodology that requires simplification or simple presentation to the reader. Thank you; to make it easier to read we have added a flow chart into Materials and Methods section.
|
|
||
|
4. Response to Comments on the Quality of English Language |
|
||
|
Point 1: |
|
||
|
Based on your suggestions, we checked again the Quality of English Language. |
|
||
|
5. Additional clarifications |
|
||
|
|
|
||
Corresponding author
Prof. Kvolik, and J. Glavas Tahtler
Round 2
Reviewer 1 Report
Work certainly is interisting since can give information about the role of Candida spp., such as posibble pathogen in lower respiratory tract. It is worrisome the intentional look for of Candida as routinary samples of urine and endotracheal aspirates since there is no breakpoint to associate the microorganism to a positive culture (no in urine neither respiratory). Then, as title mention "treatment outcome" the possible selection pressuare for to treat a Candida isolated from urine or tracheal is high and relevent since C. albicans has been enlisted in the most importan fungi resistant. So, information about this must be reviewed and reported carefully.
There is no association between title, introduction and aim. The aim is The aim of this study was to examine the frequency of any isolation of Candida spp. in the population of surgical ICU patients and the duration of mechanical ventilation compared to the control population of ICU patients. If outcome treatment in title is mentioned should be part of the main aim and a part of justification and introduction.
Line 22-23: The way as is redacted is not clear.
In abstract, acronyms or synomins must not be considered.
Line 46. Infection with Candidas is... must be Intections with Candidas are. or Infection with Candida is.
Line 59. CNS is not defined, eve we known it is central nerveous systems must be described.
Line 68: to change flora for microbiota.
Line 76: Microorganisms must be addapted to new taxonomical changes (new genus and species)
Line 94: "Rregardless of the type of sample, the number of yeasts, or whether it was infection or colonization (Figure 1). This is because samples for microbiological analysis are not routinely taken, but only from patients who have had laboratory or clinical signs of infection. " Even in those patients with infection, the search of Candida is no recommended (IDSA Laboratory 2024), but, lets asume that Candida must be followed in respiratory sample. In this context, these patients had infection with Candida spp and were treated so role as pathogen or colonizer was omitted then there is a huge bias in interpretation.
Line 179-181 To change genus and species according new nomenclature.
Table 1. Swabs from Swabs from thoracic drains, abdominal cavity, cerebrospinal fluid. I´d like to ask about the reason of swabs of CSF and the reason of isolation of Candida in this sample. Did this patient had any syntom associated with infection? Why swab? it was a craneal fracture?
Nothing commeted about treatment even is mentioned in title.

Author Response
Dear Reviewer
We carefully read your comments and corrected the manuscript according to them. We have also responded to the comments you made in the attached document.
